

# A Novel Approach for Solving CNOP and its Application in Identifying Sensitive Regions of Tropical Cyclone Adaptive Observations

Linlin Zhang[1], Bin Mu[1], Shijin Yuan[1], Feifan Zhou[2, 3]

5  [1]School of Software Engineering, Tongji University, Shanghai 201804
[2]Laboratory of Cloud-Precipitation Physics and Severe Storms, Institute of Atmospheric Physics, Chinese Academy of Sciences, Beijing 100029
[3]University of Chinese Academy of Sciences, No.19 (A) Yuquan Road, Shijingshan District, Beijing 100049

*Correspondence to:* Shijin Yuan (yuanshijin2003@163.com)

10  **Abstract.** In this paper, a novel approach is proposed for solving conditional nonlinear optimal perturbation (CNOP), named it "adaptive cooperation co-evolution of parallel particle swarm optimization and wolf search algorithm (ACPW) based on principal component analysis". Taking Fitow (2013) and Matmo (2014) as two tropical cyclone (TC) cases, CNOP solved by ACPW is used to investigate the sensitive regions identification of TC adaptive observations with the fifth-generation mesoscale model (MM5). Meanwhile, the 60 km and 120 km resolutions are adopted. The adjoint-based method (short for the ADJ-method) is also applied to solve CNOP, and the result is used as a benchmark. To validate the validity of ACPW, the CNOPs obtained from the different methods are compared in terms of the patterns, energies, similarities and simulated TC tracks with perturbations. (1) The ACPW can capture similar CNOP patterns with the ADJ-method, and the patterns of TC Fitow are more similar than TC Matmo. (2) When using the 120 km resolution, similarities between CNOPs of the ADJ-method and ACPW are higher than those using the 60 km. (3) Compared to the ADJ-method, although the CNOPs of ACPW produce lower energies, they can obtain better benefits gained from the reduction of CNOPs, not only in the entire domain but also in the sensitive regions identified. (4) The sensitive regions identified by CNOPs-ACPW has the same influence on the improvements of the TC tracks forecast skills with those identified by CNOPs-ADJ-method. (5) The ACPW has a higher efficiency than the ADJ-method. All conclusions prove that ACPW is a meaningful and effective method for solving CNOP and can be used to identify sensitive regions of TC adaptive observations.

## 1 Introduction

Tropical cyclone (TC) is one of the most frequent and influential natural hazards in the world. An accurate forecast of TC will be conducive to respond to disasters for government and people. Thus, it is essential to improve TC forecast skills. One effective way is to identify the sensitive regions of TC adaptive observations (called TCAOs for short) (Franklin and Demaria, 1992; Bergot, 1999; Aberson, 2003). Once add observations in sensitive regions-identified to reduce initial errors, better forecast will be expected (Bender et al., 1993; Zhu and Thorpe, 2006; Froude et al., 2007). Conditional nonlinear optimal



perturbation (CNOP) proposed by Mu and Duan (2003) is a nonlinear extension of the linear singular vector (SV) method, and has been applied to study sensitive regions identification of TCAOs successfully (Mu and Zhou, 2009; Qin, 2010; Zhou and Mu, 2011, 2012a, 2012b; Zhou and Zhang, 2014; Qin and Mu, 2012; Qin et al., 2013; Qin and Mu, 2014; Wang et al., 2010; Wang et al., 2013).

Comparing between the sensitive regions of CNOP-identified and the first SV-identified, Qin (2010) concludes that the former is more appropriate for TCAOs. Zhou and Mu (2011) use CNOP method to investigate the different verification areas how to affect the identification of sensitive regions. Then they study the influence of the different horizontal resolutions (2012a). Moreover, the different time and regime dependency also is researched (2012b). These research results direct the further research. Zhou and Zhang (2014) propose three schemes for identifying sensitive regions based on the CNOP method, and
recommend that the vertically integrated energy scheme. Moreover, some researchers analyse the sensitivity of dropwindsonde observations on TC predictions, which is identified by CNOP method, and conclude that the sensitive regions of CNOP-identified have a positive impact on TC track predictions (Qin and Mu, 2012; Qin et al., 2013). In studies of improvement of sensitivity-CNOP in TC intensity forecast, Qin and Mu (2014) suggest that the use of an ocean coupled model needs to be conscious, as well as the better initialization of TC vortex. Wang et al. (2013) use the CNOP method to study the mutual
affection of binary typhoons. Previous researches have proved that CNOP method is a useful and meaningful method for the above study (Zhou et al., 2013; Mu and Zhou, 2015).

Generally, there are two-type methods for solving CNOP, ones based on adjoint models (called ADJ-method for short) and ones without adjoint models. As useful and effective methods for solving CNOP without adjoint models, some modified intelligent algorithms (IAs) based on dimension reduction have been proposed and applied to solve CNOP in the Zebiak-Cane
(ZC) model successfully, such as SAEP: Simulated Annealing Based Ensemble Projecting Method (Wen et al., 2014), PPSO: principal component analysis (PCA; Jolliffe, 1986) Based Particle Swarm Optimization (Mu et al., 2015a), PCGD: Principal Components-based Great Deluge (Wen et al., 2015a), RGA: Robust PCA-Based Genetic Algorithm (Wen et al., 2015b), CTS-SS: Continuous Tabu Search Algorithm with Sine Maps and Staged Strategy (Yuan et al., 2015), and PCAGA: Principal Component Analysis Based Genetic Algorithm (Mu et al., 2015b). Compared to the ADJ-method, these methods all can obtain
CNOPs with the similar spatial patterns and acceptable objective function values, and several of them have been paralleled with the Message Passing Interface (MPI) and cost less time consumption. In the TC adaptive observation, such adjoint-free methods are also required urgently, because no having adjoint models and too high dimensions of solution space have become obstacles of solving CNOP, which is also the point of such research.

Actually, we have adopted the PCAGA method to solve CNOP for the sensitive regions identification of TCAOs with the
fifth-generation mesoscale model (MM5), and obtained meaningful results (Zhang et al., 2017). However, the resolution we used is 120 km, which is the lowest in such research. When using a higher resolution, more small-scale information can be achieved, and more accurate sensitive regions can be expected. It is necessary to use a higher resolution. Moreover, although the PCAGA achieves the meaningful results, its performance is not good enough. It is because this algorithm is based on



genetic algorithm, which has a good global searching ability but slow convergence rate. In addition, PCAGA was not parallelized in the previous study.

Therefore, in this paper, we propose a novel approach, adaptive cooperation co-evolution of PSO and wolf search algorithm (WSA) based on PCA (called ACPW for short), to solve CNOP for the sensitive regions identification of TCAOs. We take

two tropical cyclones as study cases, Fitow (2013) and Matmo (2014), and simulate them with the MM5 model using two different resolutions, 60 km and 120 km. According to the study of Zhou and Zhang (2014), we adopt the total dry energy as the objective function. And the CNOP from the ADJ-method are referred as a benchmark. To validate the validity of ACPW method, the CNOPs of the ACPW are compared with the benchmark in terms of the patterns, energies, similarities and benefits from the CNOPs reduced in the entire domain and sensitive regions identified. Further, the CNOPs with different resolutions

also are compared on these aspects. Besides, to evaluate the sensitive regions located by the ACPW, we simulate TC tracks with the initial states perturbed by the amended CNOPs in the location of the sensitive regions from those two methods. And we design two schemes to amend the CNOPs, in the same points and the equivalent proportional points of grids. In addition, we evaluate the efficiency of ACPW. All experimental results show that ACPW is a meaningful and effective method to solve CNOP for selecting the sensitive regions of TCAOs.

The organization of the paper is as follows. Section 2 describes the formalized definition of CNOP and ACPW method. In section 3 we give the design of experiments in this study. Section 4 presents the experimental analysis and results. Summaries and conclusions are provided in section 5.

## 2 Theory and Method

### 2.1 CNOP

The mathematical formalism of CNOP is described as Eq. (1). Under the constraint condition $\|u_0\|^2 \leq \delta$, an initial perturbation $\delta u_0^*$ of vector $U_0$ (initial basic state) is called CNOP, if and only if

$$J(\delta u_0^*) = \max_{\|u_0\|^2 \leq \delta} J(u_{NT}),  \tag{1}$$

where

$$u_{NT} = PM(U_0 + \delta u_0) - PM(U_0),  \tag{2}$$

and $P$ presents a local projection operator, and the value within the verification region is 1, outside is 0.

$$U_t = M(U_0, t_0, t),  \tag{3}$$

$M$ expresses a nonlinear propagation operator, $U_t$ is the development of $U_0$ at time $t$.

### 2.2 ACPW method

In this paper, we propose the ACPW to solve CNOP for identifying sensitive regions of TCAOs. The core of this approach is

the cooperation co-evolution of two intelligent algorithms: PSO and WSA, and the adaptive number of two subswarms. PSO





is a classical population based stochastic optimization technique developed by Eberhart and Kennedy (1995), inspired by social behavior of bird flocking or fish schooling, and has been applied to solve CNOP successfully and effectively in the ZC model for studying El Niño-Southern Oscillation (ENSO) predictions (Mu et al., 2015a). WSA is a new bio-inspired heuristic optimization algorithm based on wolf preying behavior, which is proposed by Rui Tang et al. (2012) and applied to study

traveling salesman problem with test functions. Their experiments show that WSA is an effective global optimizing algorithm, but needs long consuming time.

We have adopted PSO and WSA method respectively to solve CNOP in the MM5 model, but the results from them exhibit slow convergence or premature convergence. Hence, we combine the advantages of these two algorithms. We use WSA to explore in the global space due to its individuals' independence, and use PSO to dig the local space for making sure the

convergence of ACPW. Meanwhile, we design the adaptive subswarms of PSO and WSA for cooperation co-evolution. The framework of ACPW is shown in Fig. 1.

In Fig. 1, the most important part of ACPW is inside the dotted box. We divide the entire initial swarm to two subswarms with the same number of individuals, one updates individual with the PSO's rule and the other one with the WSA's rule. Then, these two subswarms are adaptively varied along with the convergence state of ACPW, i.e., when the change of the objective

function adaptive value is less than a threshold value, the number of individuals in the subswarm belonging to WSA will be increased and the other subswarm belonging to PSO will decrease the equal number of individuals to keep the same number of the whole swarm. These improvements bring better convergence accuracy and higher evolution velocity, which show in Fig. 2.

The process of solving CNOP with ACPW is described as follows:

1)  *Randomly generate an initial swarm with N individuals*. An individual $u_i$ needs to satisfy the boundary constraint in the terms of Eq. (4). Once $u_i$ goes out of the boundary, it must, thus, be pulled back, i.e.,

$$u_i = \begin{cases} u_i & \|u_i\| \le \delta \\ \frac{\delta}{\|u_i\|} \times u_i & \|u_i\| > \delta \end{cases} \quad i = 1, \cdots, N \tag{4}$$

2)  *Divide the whole initial swarm to two subswarms with an adaptive coefficient $a$*. One subswarm updates individual with the PSO' rule and the other one with the WSA' rule.

3)  *Parallelly calculate the adaptive value of the objective function, i.e. $J(u_i)$ in Eq. (1).*

4)  *Update individuals by PSO (Eq. (5)) or WSA (Eq. (6)).*

$$\begin{cases} v_i^{k+1} = \omega v_i^k + c_1 \alpha\big(o_i^k - u_i^k\big) + c_2 \beta\big(o_g^k - u_i^k\big) \\ \quad u_i^{k+1} = u_i^k + \gamma v_i^{k+1} \end{cases} \tag{5}$$

where, the superscript k or k + 1 is the iterative step, $v_i^{k+1}$ is the velocity of the individual $u_i^k$ and calculated by the first subformula. $\omega$ is the inertia coefficient, $c_1$ and $c_2$ are the learning factors, $\alpha$ and $\beta$ are the random numbers uniformly

distributing on the interval from 0 to 1. $o_i^k$ is the local optimum and $o_g^k$ is the global optimum in the $k^{th}$ iteration. $\gamma$ is the restraint factor to control the speed. $u_i^{k+1}$ is the updated individual with PSO.



There are two ways for updating individual in WSA, prey and escape, which represent the functions of searching in a local region and escaping from a local optimum.

$$\begin{cases} u_i^{k+1} = u_i^k + \theta \cdot r \cdot rand(\ ) & dist\left(u_i^k, u_i^{k+1}\right) < r. and. J(u_i^k) < J(u_i^{k+1}) \\ u_i^{k+1} = u_i^k + \theta \cdot s \cdot escape(\ ) & p > p_a \end{cases} \tag{6}$$

where the superscript k or k + 1 is also the iterative step, $\theta$ is the velocity, $r$ is the local optimizing radius, which smaller than the global constraint radius $\delta$. $rand(\ )$ is the random function, whose mean value distributed in [-1,1]. $escape(\ )$ is the function of calculating a random position, which is larger 3 times than $r$. $s$ is the step size of the updating individual. $p$ is a random number in [0,1], $p_a$ is the probability of individual escaping from the current position.

5) *Judge whether the change of the adaptive value of the objective function is smaller than ε.* If yes, set a new value to the adaptive subswarm coefficient $a$. If not, continue running the process.

6) *Judge whether the termination condition is satisfied*. If yes, terminate the iteration. Otherwise, go to step 2.

All above processes are based on the dimension reduction with PCA, whose procedure has been described in the study of Mu et al. (2015a). After many experiments, the parameters of ACPW are set in Table. 1.

Although the parameters are more than each single algorithm, but most of them still is the same empirical value of each algorithm, which need not to adjust. The reason of using different number of individuals is that the memory of internal storage we used is not enough when using more than 200 individuals and ACPW will be interrupted.

## 3 Experiments Design

All the experiments are run on a Lenove Thinkserver RD430 with two Intel Xeon E5-2450 2.10 GHz CPUs, 32 logical cores and 132G RAM. And the operating system is CentOS 6.5. All the codes are written in FORTRAN language and compiled by PGI Compiler 10.2.

### 3.1 The model and Data

In this paper, we adopt the MM5 model to study the sensitive region identification of TCAOs, and the corresponding adjoint system of MM5 model (Zou et al., 1997) is used to obtain the benchmark. And the ERA interim daily analysis data (1 °×1 °) (Dee et al., 2011) from the European Center for Medium range Weather Forecasts (ECMWF) are used to generate the initial conditions and boundary conditions. The physical parameterization scheme is constructed as follows: dry convective adjustment, the high-resolution planetary boundary layer scheme, grid resolved large-scale precipitation and the Kuo cumulus parameterization scheme.

We also utilise the best TC track data (Ying et al., 2014) from the China Meteorological Administration - Shanghai Typhoon Institute (CMA-SHTI) as TC tracks observed for evaluating the simulation TC tracks of the MM5 model.



### 3.2 Typhoons synop to Fitow (2013) and Matmo (2014)

TC Fitow (2013) and TC Matom (2014) are taken as the study cases and introduced below. Fitow is the 23st TC in 2013, and develops to the east of the Philippines on September 29 and strikes China at Fuding in Fujian province On October 6. Matom is the 10th named typhoon in 2014, and it happens on July 17 and lands Taiwan on July 22. For these two cases, 24-h control forecasts are set as background fields which integrate from 0000 UTC 5 Oct 2013 to 0000 UTC 6 Oct 2013 (TC Fitow) and from 1800 UTC 21 Jul 2014 to 1800 UTC 22Jul 2014 (TC Matom). After the 24h period, TC Fitow has the maximum sustained wind of 162 kilometres per hour and TC Matmo has 151.2 kilometres per hour. In addition, the forecasts are executed at the 60 km and 120 km resolutions with 11 vertical levels and the model domain covers 55×55, 21×26 grids, respectively.

The simulated TC tracks of MM5 model for these two cases are acceptable, as has been shown in our previous study (Zhang et al. 2017). The following work will base on those simulations.

### 3.3 Experimental setup

Depending on the conclusion that a little change of the verification areas never hurts the results (Zhou and Mu, 2011), we design the verification areas as rectangles covering the potential typhoon tracks at forecast time.

The initial perturbation sample $\delta u_0$ is composed of the perturbed zonal wind $u'_0$, meridional wind $v'_0$, temperature $T'_0$ and surface pressure $p_{s_0}'$. Each component can be represented as a $m \times n \times l$ matrix. $m \times n$ is the distribution of the horizontal grid and $l$ denotes the number of vertical levels. In order to extract features for reducing the dimensions of solving CNOP, the $m \times n \times l$ matrix is reshaped to a $k \times 1$ vector, here $k = m \times n \times l \times S$ (S is the number of the components). Assuming we have R vectors to represent the features of the solution space, we recombine the R vectors to a $k \times R$ matrix, and use PCA to capture the feature space with lower dimensions. Then the CNOP is solved in the feature space until obtain the global CNOP, which will be projected to the original solution space. When using the ACPW to solve CNOP, its initial inputs are produced by the random way in the feature space, and the CNOP has the largest nonlinear evolution at prediction time, i.e. the largest adaptive value of the objective function in Eq. (8).

There exists

$$f(i, j) = \int_0^1 E_T(i, j, \sigma) \, d\sigma \, , \tag{7}$$

where $E_T(i, j, \sigma)$ denotes the total dry energy of CNOP at the MM5 grid point $(i, j, \sigma)$.

Corresponding to formula (1) and (2), we have

$$(u_{NT}) = \frac{1}{D} \int_D \int_0^1 \left[ u_t'^2 + v_t'^2 + \frac{c_p}{T_r} T_t'^2 + R_a T_r \left( \frac{p_{st}'}{p_r} \right)^2 \right] d\sigma dD, \tag{8}$$

where $u_t'$, $v_t'$, $T_t'$, $p_{s_t}'$ are components of $u_{NT}$, which is the nonlinear development of perturbed $U_0$ (i.e. $U_0 + \delta u_0$) from the initial time $t_0$ to the prediction time $t$. $\sigma$ is the vertical coordinate. And Table. 2 illustrates the other reference parameters.

For the convenience of optimization, solving CNOP can be transformed to a minimum problem, as follows:



$$J(\delta u_0^*) = \min_{\|u_0\|^2 \le \delta}\left(-\frac{1}{D}\int_D \int_0^1 \left[u_t'^2 + v_t'^2 + \frac{c_p}{T_r}T_t'^2 + R_a T_r\left(\frac{p_{st}'}{p_r}\right)^2\right]d\sigma dD\right),\qquad(9)$$

To facilitate understanding, all symbols are listed in Table 2 and their meanings are also explained.

## 4. Experimental Results and Analysis

To validate the validity of ACPW, we compare CNOPs-ADJ-method and CNOPs-ACPW on the CNOP patterns, energies,
similarities, benefits from reduction of CNOPs, and simulated TC tracks with perturbations.

### 4.1 CNOP patterns

In this subsection, we compare CNOPs obtained from the ADJ-method and ACPW on the patterns of temperature and wind.
Experimental results show that TC Fitow has more similar CNOP patterns than TC Matmo. The CNOP patterns are described
in Fig. 3.

To the 120 km resolution for TC Fitow (Fig. 3a, b), these two methods have almost same major warm locations and similar
cold parts, while the wind vectors have opposite directions. The ADJ-method catches the CNOP with two major locations. The
red (warm) one distributes at the west of the initial cyclone (called IC for short), and the green (cold) one distributes at the
north of the IC. The ACPW also captures the CNOP with two main locations. The warm one distributes at the west and the
cold one locates at the northwest of the IC. In this subsection, the spatial orientation is all relative to the position of the IC.

Therefore, in the following part, we explain the spatial orientation in the figures without repeating the IC.

For TC Fitow with the 60 km resolution (Fig. 3c, d), the CNOP spatial distribution of the ACPW is very similar to the ADJ-
method's. At the northwest in the verification area, two CNOPs have two similar major parts, one warm area and one cold
area. The difference between these two patterns is that, the ADJ-method has another major warm area locating at the northwest
and the ACPW has another major warm area locating at the east. Besides, the distribution of secondary parts has a slight

difference.

For the same method with the different resolution (Fig. 3a, c and Fig. 3b, d), the CNOP patterns have similar major distributions
at the northwest, but with a different region. The reason is when using a higher resolution, more small-scaled things will be
resolved (Zhou and Mu, 2012a).

For TC Matmo with the 120 km resolution (Fig. 4a, b), the ADJ-method and ACPW obtain CNOPs with different spatial

patterns of temperature and wind. The ADJ-method has two major parts with the warm one locating at the west and the cold
one distributing at the east. The ACPW has two main parts distributing at the northeast with one warm area near to the IC and
cold one far from the IC. For TC Matmo with the 60 km resolution (Fig. 4c, d), in the verification area, these two CNOP
patterns have similar spatial distributions with two warm areas locating at the same positions almost. But the parts outside of
the verification area are distributed in the different positions. Besides, the CNOP of ADJ-method has more regular distributions



than the ACPW's. For the same method with the different resolution (Fig. 4a, c and Fig. 4b, d), the CNOP patterns cover similar rough areas but with different ranges and details.

Based on the above analysis about patterns of temperature and wind, we can conclude that, when using the 60 km resolution, the CNOPs of ADJ-method and ACPW have more similar major patterns than those with the 120 km. In addition, ACPW can

obtain CNOPs with similar patterns in TC Fitow than in TC Matmo.

Vertically integrated energies of CNOPs for TC Fitow are displayed in Fig. 5. Compared to the ADJ-method, when using the 120 km resolution, the CNOP of ACPW has much lower energy and various positions, but when using 60 km resolution, it can get similar energies and positions. Besides, the energy of CNOP obtained from ACPW has a larger range in the center.

Vertically integrated energies of CNOPs for TC Matmo are displayed in Fig. 6. Compared with the ADJ-method, when using the 120 km resolution, the CNOP of ACPW has a lower energy and covers large areas, but when using the 60 km resolution, although its energy is still lower, but positions are getting closer.

## 4.2 Similarities

When we evaluate CNOPs, in addition to characteristic and distributions of CNOP patterns, consideration should also be given

to numerical similarities and to the benefits from CNOPs. Therefore, we calculate the similarity between CNOP-ADJ-method and CNOP-ACPW, and use X and Y to represent them in the formula (10).

$$S_{xy} = \frac{\langle X,Y \rangle}{\sqrt{\langle X,X \rangle}\sqrt{\langle Y,Y \rangle}}, \qquad (10)$$

The results show in Table. 3. The similarity values can reflect the similarities among the patterns of CNOPs (Fig. 3 and Fig. 4).

In Table. 3, for TC Fitow, the similarity with the 120 km is -0.83, and that with the 60 km resolution is 0.43. For TC Matmo, the similarity with the 120 km resolution is 0.42, and that with the 60 km resolution is 0.37. The negative sign represents that parts of the CNOPs of these two methods have opposite wind vector directions, which showed in Fig. 3. We also find that when using a higher resolution, the similarity is lower. The reason is that although the major patterns of those CNOPs are similar, the other secondary parts of them are different, and they cover larger areas. Actually, when using a higher resolution,

we can achieve more small-scale information and the sensitive regions identification will be more accurate. As the analysis of CNOP patterns, we assuredly get more similar major patterns when using the 60 km resolution than using the 120 km resolution, but compared with the other different parts, the similar parts are very small. However, the similarities decreased will not affect identifying the sensitive regions, because the adaptive observation only focuses on the points with bigger influence, which will be proved in subsection 4.4 of this section.

We also compare the energy for 24 hours nonlinear developments under the initial states perturbed by different CNOPs, i.e. $J(M(U_0 + \delta u_0^*))$. The results are shown in Table. 4.





Results show that all CNOPs obtained using the ACPW produce lower energies than the ADJ-method, but when reducing CNOPs to W×CNOPs in the entire domain and reducing CNOPs to 0.5 time in the sensitive regions identified, the ACPW can obtain better benefits which will be discussed in following subsection.

### 4.3 Benefits from Reduction of CNOPs

In this subsection, we design two groups of idealized experiments to investigate the validity of sensitive regions identified by CNOPs, based on two assumptions that:

When adding adaptive observations in sensitive regions identified, the environment around is idealized, and the improvements of observations added are reducing original errors to 0.5 times.

CNOPs achieved by us can be seen as the optimal initial perturbations. Once we reduce them in the sensitive regions, the

benefits earned will be the best.

Under the above assumptions, reducing CNOPs to W×CNOPs and inserting them to the initial states can investigate the reductions of CNOPs how to influence TC forecast skills. Besides, reducing the values of CNOPs to 0.5 time in the sensitive regions identified by vertically integrated energies can investigate that adding adaptive observations in the sensitive regions how to impact on TC forecast skills.

First, as CNOP can be seen as the optimal initial perturbations in the TCAOs, we reduce CNOP to W×CNOP, W is a coefficient in (0, 1), and insert CNOP-reduced into the initial state with 24-h evolution of the nonlinear model of the MM5 model, then calculate the forecast error with formula (11) to gain benefits from such reductions. Second, we determine the sensitive regions with vertically integrated energies using two schemes: the same points of energies in the different resolutions, and the equivalent percentage of points of the different grids. Then we reduce CNOPs to 0.5 time only in the sensitive regions

and insert CNOPs-amended to the initial states with 24-h evolution of the nonlinear model. The experimental results are denoted below.

### 4.3.1 Reducing CNOP to W×CNOPs in the entire domain

In this part, we explore the forecast improvement extents of reducing CNOPs to W×CNOPs in the entire domain. The scheme is inserting CNOP-reduced into the initial state with 24-h evolution of the nonlinear model of the MM5 model. The prediction

error is computed by the formula (11):

$$J_1(u_{NT}) = \|PM(U_0 + \delta u_0) - PM(U_0)\|^2, \tag{11}$$

Where the definitions of $u_{NT}$, P, M and $U_0$ are the same with those in Eq. (1), (2) and (3).

And the prediction error after reducing CNOP in the entire domain is computed by the formula (12):

$$J_2(u_{NT}) = \|PM(U_0 + W\delta u_0) - PM(U_0)\|^2, \tag{12}$$

where W is the weighting coefficient, and it is set as 0.25, 0.5 or 0.75 for decreasing error. And the benefit from such reductions is calculated by the formula (13):





$$\frac{J_1(u_{NT}) - J_2(u_{NT})}{J_1(u_{NT})}, \tag{13}$$

Obviously, the prediction benefit is increasing when W gets decreasing. Fig. 7 and Fig. 8 also show that the ACPW can obtain CNOPs with better benefits of reducing CNOPs to W×CNOPs in the entire domain than the ADJ-method, except for the W is 0.25 for TC Fitow with the 60 km resolution. The reason is that the ACPW optimizes in a low-dimensional feature space from

PCA, and focuses on more effective points in the entire domain, which has positive effects on improving the forecast.

**4.3.2 Reducing CNOP to 0.5 time in the sensitive** regions

In this part, we explore the forecast improvement extent which gained from reducing CNOPs to 0.5 time in the sensitive regions. We determine the sensitive regions with vertically integrated energies using two schemes: the 20 biggest points of energy in the different resolution, and the 1/100 points of the different grids, which is 30 points in the 60 km resolution (55×55)

and 6 points in the 120 km resolution (21×26). The sensitive regions with the 20 biggest points of energy are denoted in Fig. 9 and Fig. 10.

In Fig. 9 and Fig. 10, we can see that when the equivalent points are adopted, a bigger scope is covered with the 120 km resolution than with the 60 km resolution. When using the 20 points from the ADJ-method and ACPW as the sensitive regions and reducing CNOPs to 0.5 time in these points, the benefits are displayed in Table. 5.

In Table. 5, for TC Fitow, compared to the ADJ-method, i.e. 5.93% in the 120 km resolution and 3% in the 60 km resolution, ACPW obtains a higher benefit (8.05%) in the 120 km resolution, and a lower benefit (-0.84%) in the 60 km resolution. -0.84% means the reduction of CNOP cannot obtain a benefit, but narrows the quality of the initial state. For TC Matmo, the ACPW achieves a much higher benefit (20.48%) than the ADJ-method's (6.12%) in the 60 km resolution, while a lower benefit (16.26%) than the ADJ-method's (20.90%) in the 120 km resolution. In addition, when using the same number of energy

points, the benefits in the 120 km resolution are almost higher than those in the 60 km resolution, except for the benefit of ACPW in the 60 km resolution for TC Matmo.

The sensitive regions with the 1/100 points of the different grids are denoted in Fig. 11 and Fig. 12.

Fig. 11 and Fig. 12 shows that when using the different resolutions, the sensitive regions identified by the same method are different. And the sensitive regions identified by the ACPW are more dispersive than those identified by the ADJ-method,

which is attributed to randomness of intelligent algorithms. Table. 6 shows the benefits gained from reducing CNOPs to 0.5 time in the sensitive regions identified by the ADJ-method and ACPW with different points in the different resolutions.

In Table. 6, for TC Fitow, the ACPW achieves a 4.23% benefit, which is higher than the ADJ-method (3.9%) in the 60 km resolution, and a lower benefit 0.01% than the ADJ-method (1.72%) in the 120 km resolution. For TC Matmo, the ACPW also has a higher benefit (9.75%) and a lower benefit (6.86%) than the ADJ-method (1.21% and 13.24% respectively).

Combined with Table. 5 and Table. 6, we can conclude that the sensitive regions cover bigger scope, higher benefits will be obtained. When using the same proportion of grids with the different resolutions, the sensitive regions under the higher





resolutions will achieve the higher benefits. These results also prove that CNOPs obtained from the ACPW can identify sensitive regions with higher benefits in the 60 km resolution.

## 4.4 Simulated TC Tracks

To investigate the validity of the sensitive regions identified by CNOP further, we compare simulated TC tracks of the MM5

model for each case with inserting CNOPs or W×CNOPs into the initial states, and also simulate TC tracks with inserting CNOPs-amended in the different sensitive regions (20 or 30 points). As 120 km is the lowest resolution in such research, and the tracks cannot be drawn under this resolution in our study, we only analyse the simulated TC tracks with the 60 km resolution. To demonstrate clearly, we draw two tracks in a subfigure, which are observed TC track from the CMA-SHTI and simulated TC track from the MM5 model with overlaying the different perturbations onto the same initial states. According to the

experimental results, when overlaying the CNOPs or amended CNOPs onto the same initial states, although the CNOPs are obtained from the different methods, the simulated tracks are the same. Therefore, we only exhibit one group of figures for each case. The results are presented in Fig. 13 and Fig. 14.

Fig. 13 demonstrates the simulated TC tracks of the MM5 with inserting CNOP or W×CNOP into the initial state for TC Fitow and four subfigures are the same. The reason is that the deviations between the simulated TC and observed TC track is very

small, it is not easy to make improvements. Hence, when inserting different CNOPs into the identical initial states to simulate TC tracks, the change is not evident. Besides, the resolution we used is the 60 km, which is not high enough to show more details about changing tracks.

Fig. 14 demonstrates the simulated TC tracks of MM5 with inserting CNOP or W×CNOP into the initial state for TC Matmo. Subfigure (a) and (b) are the same, and from (b) to (d), the simulated positions after 24 hours are getting closer to the observed

positions. It illustrates that when CNOPs achieved by the ACPW and ADJ-method being seen as the optimal initial perturbations, reducing CNOPs have positive effects on the forecast skills of the simulated tracks. And that also proves the ACPW is a meaningful and effective method for solving the approximate CNOP of the ADJ-method.

We also simulate TC tracks with inserting the amended CNOPs, which are reduced to 0.5 time only in the sensitive regions. And we use 20 and 30 points as the sensitive regions to study such difference how to affect the forecast skills. And the results

are shown in Fig. 15 and Fig. 16.

In Fig. 15 and Fig. 16, the simulated TC tracks are the same, not only the different method but also the different sensitive regions. We can conclude that the ACPW, an adjoint-free method, is a meaningful and effective method for solving the approximate CNOP of the ADJ-method. According to these results, we also conclude that using 20 or 30 points as the sensitive regions, the same improvements are achieved in the TC tracks forecast skills, so that we can use fewer points in the real

adaptive observations to reduce costs.



## 4.5 The efficiency of ACPW

To promote the efficiency of ACPW, we parallel it with MPI technology. The time consumption of each case is the same almost. Hence, we can use one group of experimental results to elucidate the efficiency of ACPW. Since the ADJ-method cannot be parallelized because of its each input depending on the output of the previous step, its time consumption is not

changed. And as this method generally uses 4~8 initial guess fields to obtain the optimal value, we use one and four initial first guess fields to achieve CNOP. The time consumption of the ADJ-method and ACPW is shown in Table. 7.

1. ADJ-method (1) means using 1 initial guess field, and ADJ-method (1) means using 4 initial guess fields.

When using the 120 km resolution, the time consumption of ADJ-method using 1 and 4 initial guess fields is 12.4 minutes, 49.7 minutes respectively. And when using the 60 km resolution, the time consumption is 79.9 minutes, 321.1 minutes

respectively. Unlike the ADJ-method, the ACPW has been paralleled, and when using 22 cores, the ACPW costs much less time, 2.74 minutes for the 120 km resolution and 20.8 minutes for the 60 km resolution. Obviously, the ACPW has higher efficiency when using the different resolutions. Compared to the ADJ-method (1), the speedup reaches 4.53 and 3.84 for the different resolutions. Compared to the ADJ-method (4), the speedup reaches 18.14 and 15.44.

## 5 Summaries and Conclusions

In this study, we present a novel approach, adaptive cooperation co-evolution of paralleled PSO and WSA (ACPW), to solve CNOP. And the CNOP based on the ACPW is applied to study sensitive regions identification of TCAOs in the MM5 model, without using the adjoint model. We study two TC cases, Fitow (2013) and Matmo (2014), with 60 km and 120 km resolutions. The objective function is set as the total dry energy, which is the 24 hours nonlinear development of initial perturbations at the prediction time within verification area. We also calculate CNOP with the ADJ-method and the result is seen as a benchmark.

To validate the validity of ACPW, the CNOPs obtained from the different methods are compared in terms of the patterns, energies, similarities, benefits of reduction of CNOPs and simulated TC tracks with perturbations.

 According to all experiments, we can get five conclusions as follows:

(1) Compared with the ADJ-method, the ACPW can obtain CNOPs with more similar patterns of temperature and wind for TC Fitow than those for TC Matmo.

(2) When using the 120 km resolution, the similarities of CNOPs achieved by the ADJ-method and ACPW are higher than those using the 60 km. The reason is that although the major patterns of those CNOPs are similar, the other parts of them are different, which cover larger areas. Actually, when using a higher resolution, we can achieve more small-scale information and sensitive regions identification will be more accurate. As the analysis of CNOP patterns, we assuredly get more similar major patterns when using the 60 km resolution than using the 120 km resolution, but the similar parts are very small compared

with the other different parts. However, the decreased similarities will not affect identifying sensitive regions, because the adaptive observation only focuses on the points with bigger influence.



(3) Under the assumptions that when adding adaptive observations in the sensitive regions identified, the environments around is idealized, and the improvements of observations added are reducing original errors to 0.5 time; CNOPs achieved by us can be seen as the optimal initial perturbations, once we reduce them in the sensitive regions, the benefits earned will be the best. We design two groups of idealized experiments to investigate the validity of sensitive regions identified by CNOPs for TC

tracks forecast skills: reducing CNOPs to W×CNOPs and reducing the values of CNOPs to 0.5 time in the sensitive regions identified using the vertically integrated energies. The experimental results show that the CNOPs of ACPW produce lower energies than the ADJ-method, but can obtain better benefits when reducing CNOPs in the above two ways.

(4) The ACPW can gain the effective CNOPs for identifying the sensitive regions, which have the same influences on the forecast improvements of the simulated TC tracks with the ADJ-method. We compare the different forecast improvements of

the TC tracks earned from the different perturbations reduced, including reducing CNOPs to W×CNOPs in the entire domain and reducing CNOPs to 0.5 time in the sensitive regions. The experimental results all support our conclusions.

(5) The ACPW has a higher efficiency than the ADJ-method. Compared to the ADJ-method using 1 initial guess field, the speedup reaches 4.53 for the 120 km resolution and 3.84 for the 60 km resolution. Compared to the ADJ-method using 4 initial guess fields, the speedup reaches 18.14 and 15.44, respectively.

All conclusions prove that ACPW is a meaningful and effective method for solving approximate CNOP in identifying sensitive regions of TCAOs. In addition, as we reduce the dimensions with PCA, CNOPs obtained by us will lose some energies. Compared to the CNOPs-ADJ-method, the CNOPs-ACPW all are local CNOPs. But in the ACPW, they are the global CNOPs. Since PCA makes our optimization focusing on more effective points with higher energies, that the ACPW can achieve the CNOPs bringing the better benefits and the same influence on the improvements of the TC tracks forecast skills.

We are restricted to computation sources for the time being, which also limits the parallelization of ACPW. We will improve the computation conditions, and use parallel ACPW to solve CNOP in the weather research forecast (WRF) model with a finer grid and higher resolutions. In addition, we will apply this-type method to solve CNOP in the community earth system model (CESM) model, which does not have an adjoint model.

**Acknowledgments:** In this paper, the research was sponsored by the Foundation of National Natural Science Fund of China
(No.41405097).

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




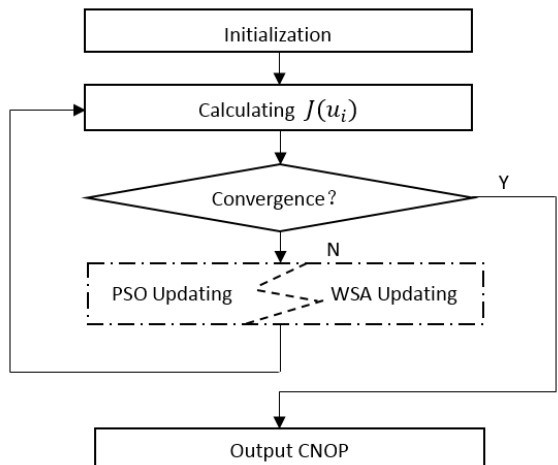

**Figure 1: The framework of ACPW method.**

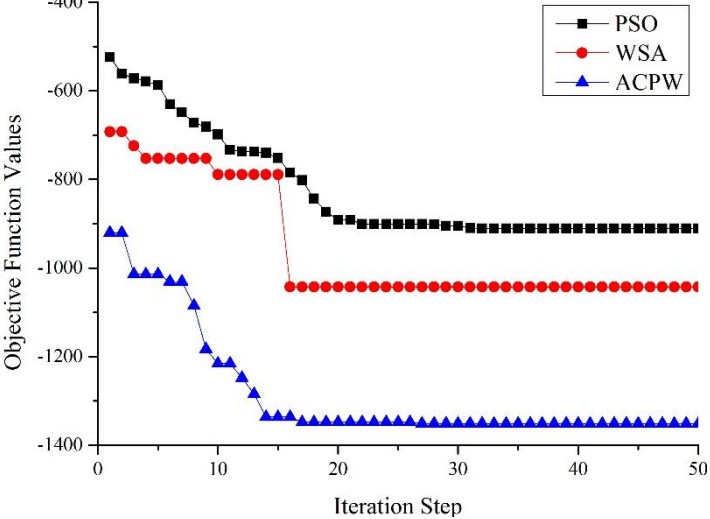

**Figure 2: Convergence of PSO, WSA and ACPW. PSO is denoted as the black line with squares, WSA is shown as the red line with**
5 **circles and ACPW is represented as the blue line with triangles.**




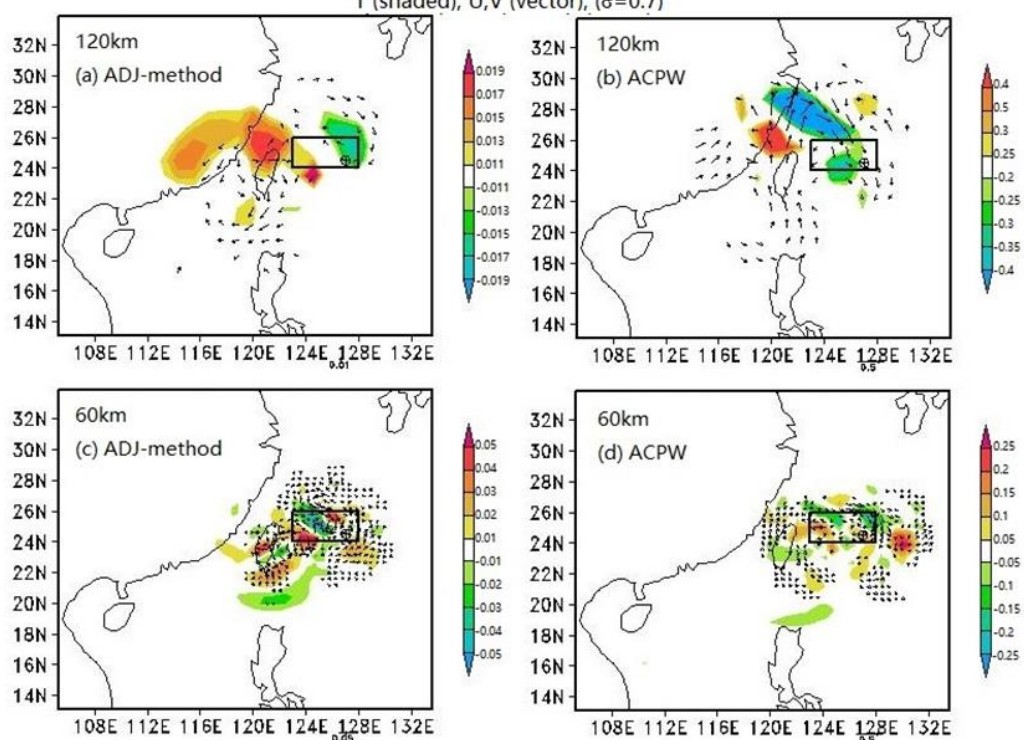

**Figure 3: CNOP patterns at σ=0.7 for TC Fitow. The shaded parts represent the temperature (units: K) and the vectors describe the wind (units: m s-1). The squares draw the verification areas and the initial cyclone positions are shown on ⊕. (a) and (b) denote the CNOP patterns with 120 km resolution of ADJ-method and ACPW, respectively; (c) and (d) represent the CNOP patterns with 60 km resolution of ADJ-method and ACPW, respectively.**




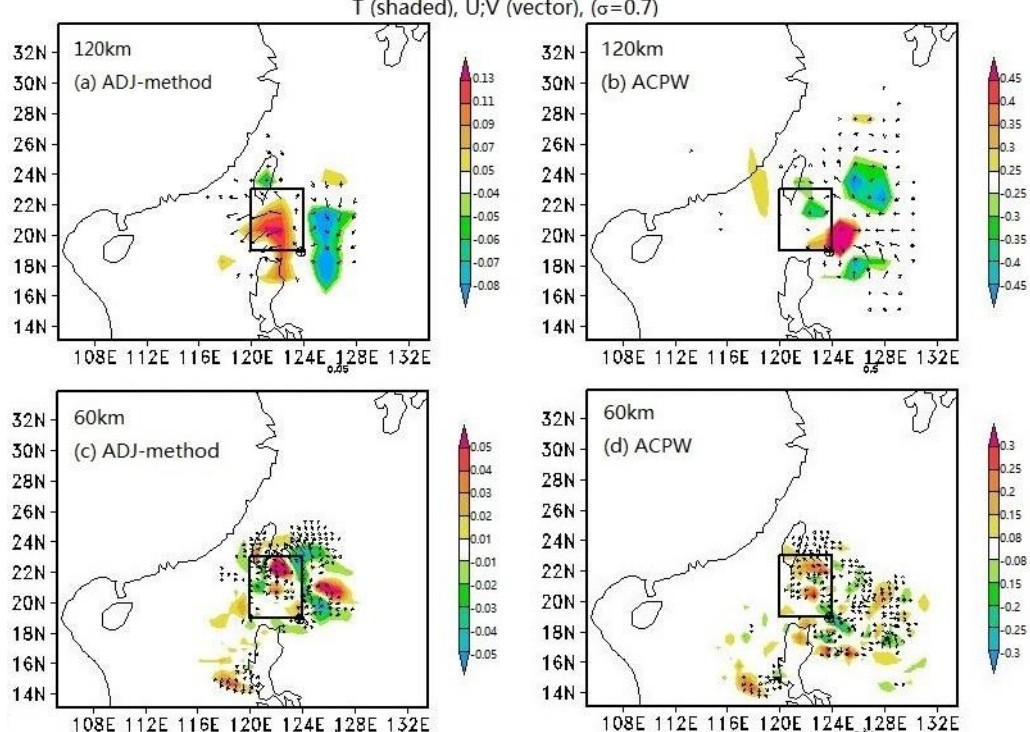

**Figure 4: As in Fig. 3 except for tropical storm Matmo. The shaded parts represent the temperature (units: K) and the vectors describe the wind (units: m s-1). The squares draw the verification areas and the initial cyclone positions are shown on ⊕. (a) and (b) denote the CNOP patterns with 120 km resolution of ADJ-method and ACPW, respectively; (c) and (d) represent the CNOP patterns with 60 km resolution of ADJ-method and ACPW, respectively.**





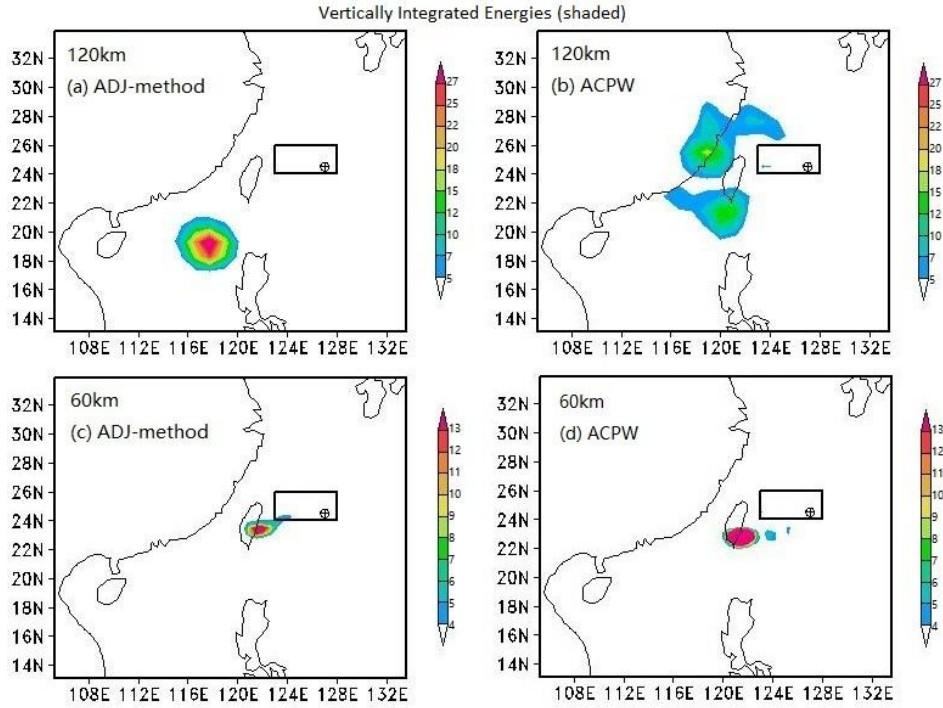

Figure 5: Same as Fig. 3, but the shaded parts represent the vertically integrated energies (units: J kg-1).

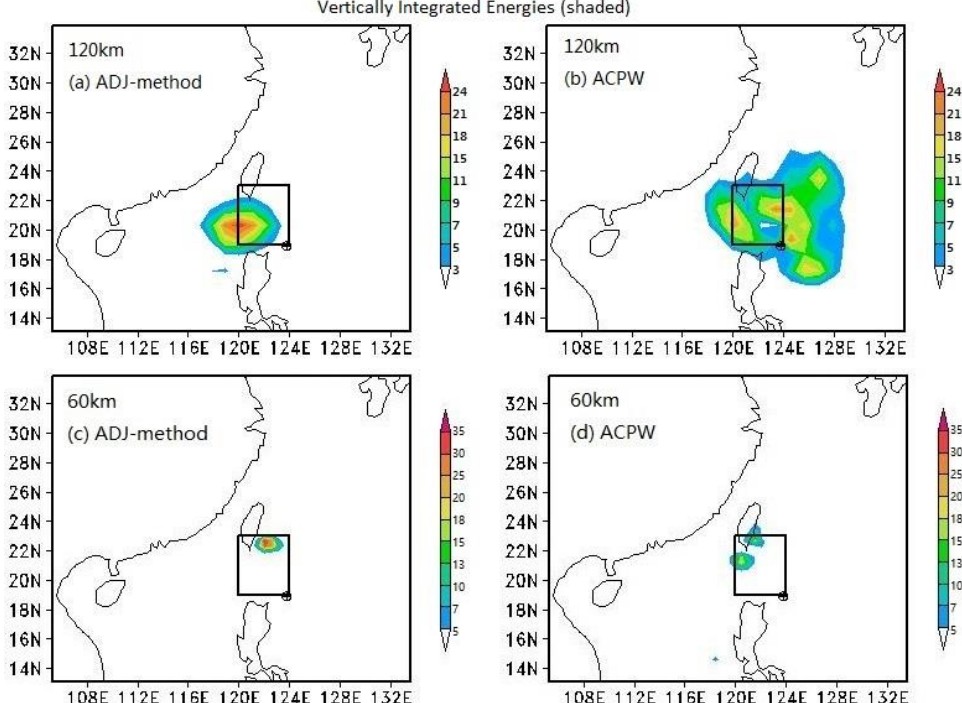

Figure 6: Same as Fig. 4, but the shaded parts represent the vertically integrated energies (units: J kg-1)

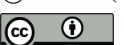



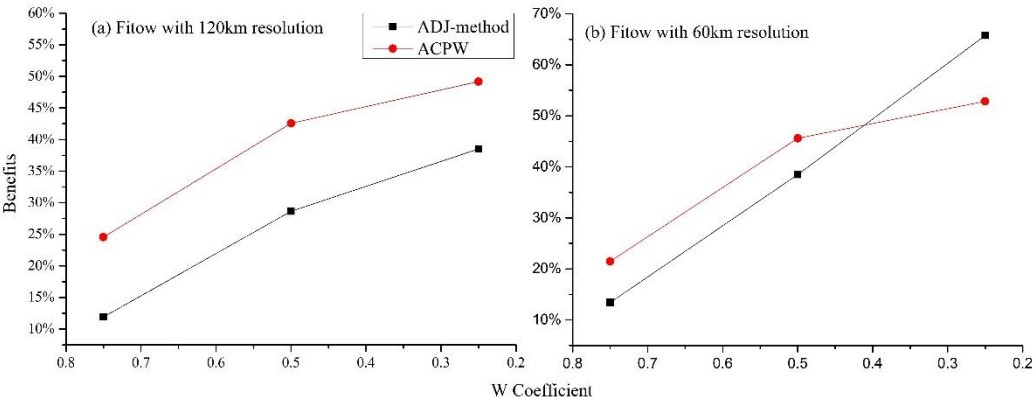

**Figure 7: Benefits (percent, %) gained from reducing CNOPs to W×CNOPs achieved by ADJ-method and ACPW in the entire domain for TC Fitow (2013). The x-coordinate is the W coefficient values. And the y-coordinate denotes the benefits (percent, %) derived from the two methods. ADJ-method is described as black line with squares and ACPW is red line with circles.**

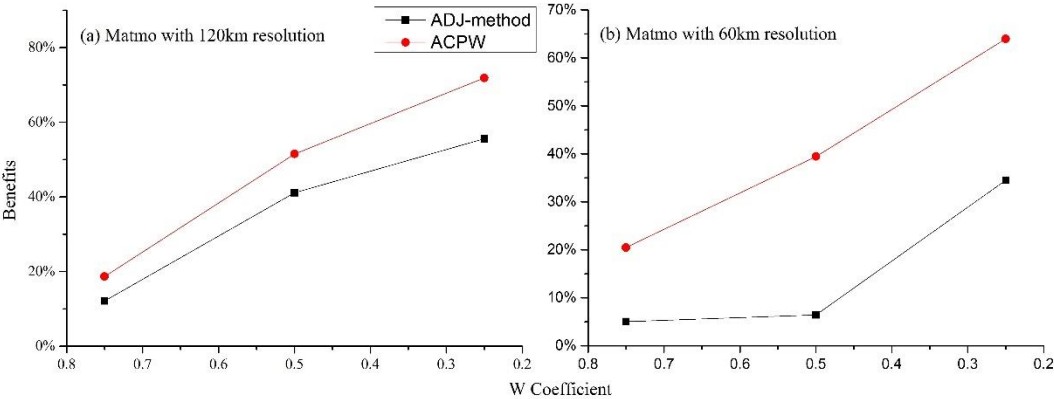

**Figure 8: Benefits (percent, %) gained from reducing CNOPs to W×CNOPs identified by ADJ-method and ACPW in the entire domain for TC Matmo (2014). The x-coordinate is the W coefficient values. And the y-coordinate denotes the benefits (percent, %) derived from the two methods. ADJ-method is described as black line with squares and ACPW is red line with circles.**



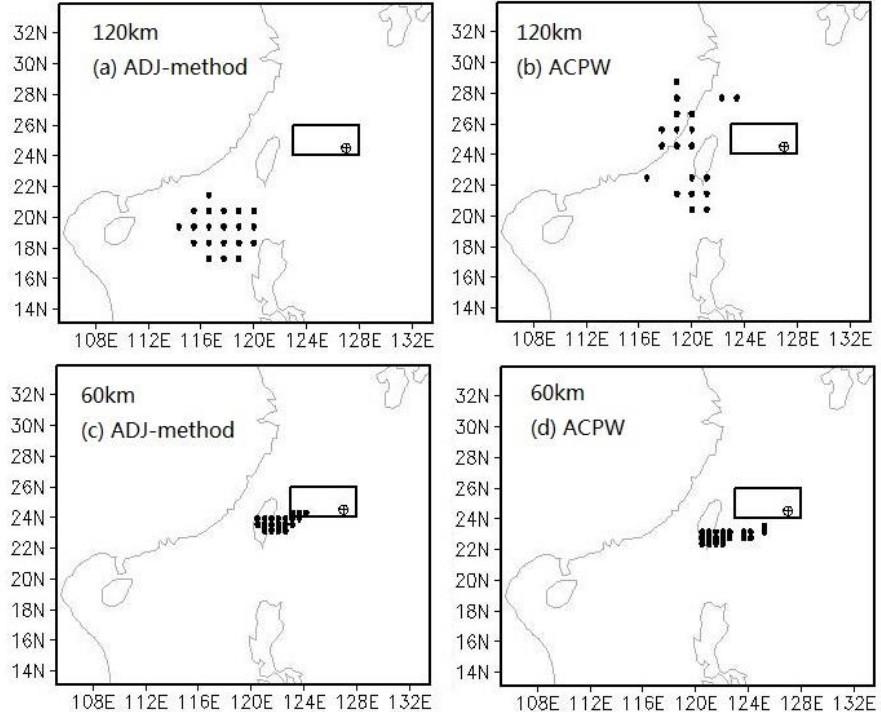

**Figure 9: Sensitive regions identified by CNOPs with 20 points for TC Fitow. The squares draw the verification areas and the initial cyclone positions are shown on ⊕. (a) and (b) denote the CNOP patterns with 120 km resolution of ADJ-method and ACPW, respectively; (c) and (d) represent the CNOP patterns with 60 km resolution of ADJ-method and ACPW, respectively.**





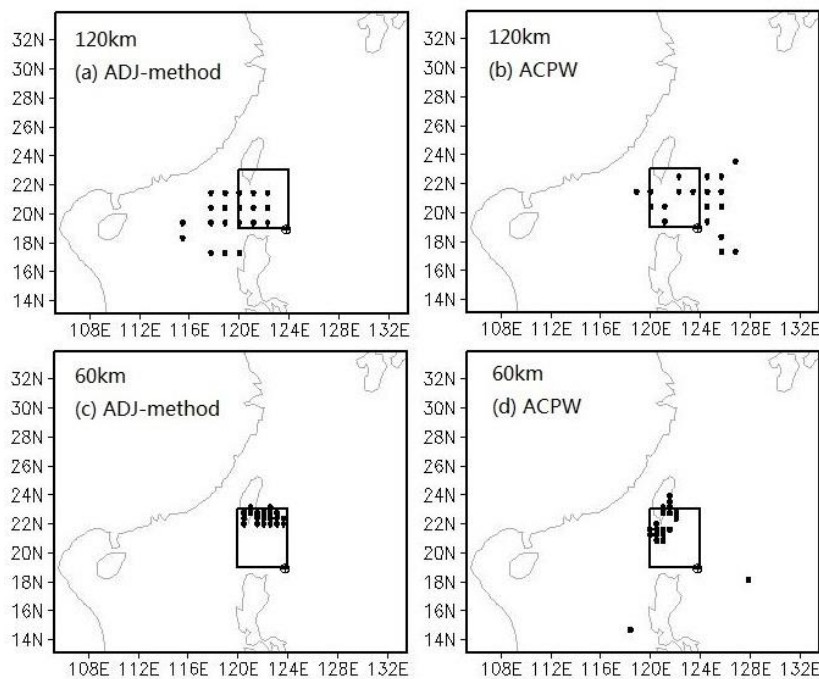

**Figure 10: Sensitive regions identified by CNOPs with 20 points for TC Matmo. The squares draw the verification areas and the initial cyclone positions are shown on ⊕. (a) and (b) denote the CNOP patterns with 120 km resolution of ADJ-method and ACPW, respectively; (c) and (d) represent the CNOP patterns with 60 km resolution of ADJ-method and ACPW, respectively.**

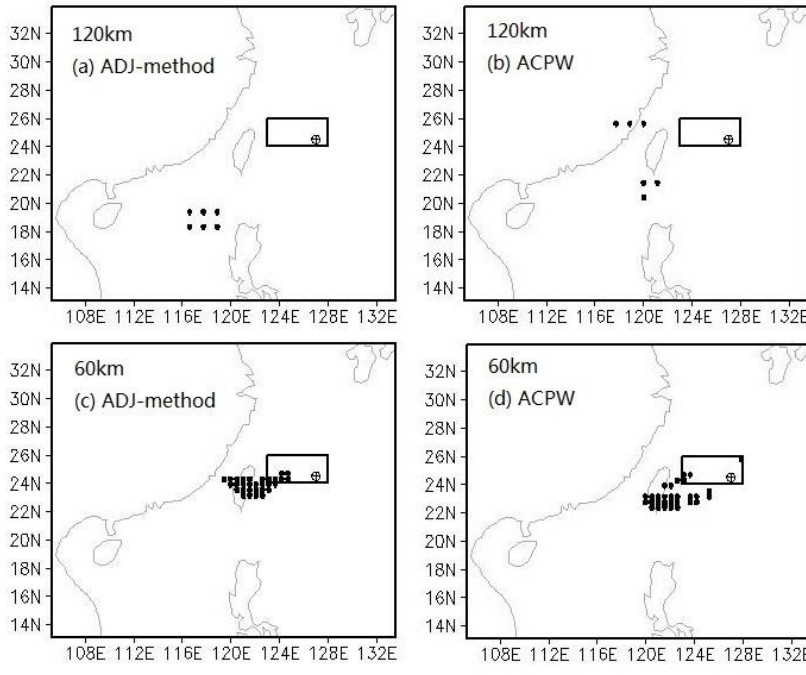




**Figure 11: Sensitive regions identified by CNOPs with 6 points in 120 km resolution and 30 points in 60 km resolution for TC Fitow. The squares draw the verification areas and the initial cyclone positions are shown on ⊕. (a) and (b) denote the CNOP patterns with 120 km resolution of ADJ-method and ACPW, respectively; (c) and (d) represent the CNOP patterns with 60 km resolution of ADJ-method and ACPW, respectively.**

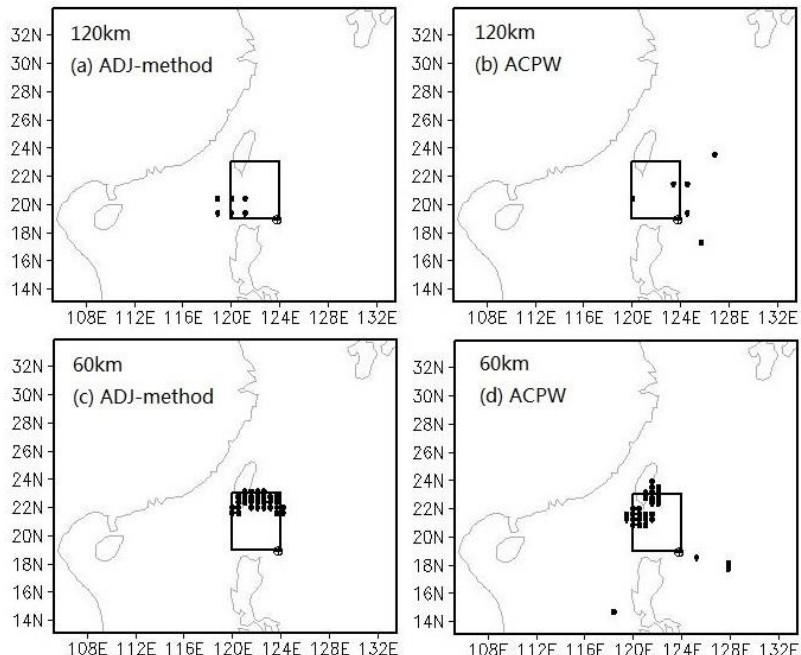

**Figure 12: Sensitive regions identified by CNOPs with 6 points in 120 km resolution and 30 points in 60 km resolution for TC Fitow. The squares draw the verification areas and the initial cyclone positions are shown on ⊕. (a) and (b) denote the CNOP patterns with 120 km resolution of ADJ-method and ACPW, respectively; (c) and (d) represent the CNOP patterns with 60 km resolution of ADJ-method and ACPW, respectively.**



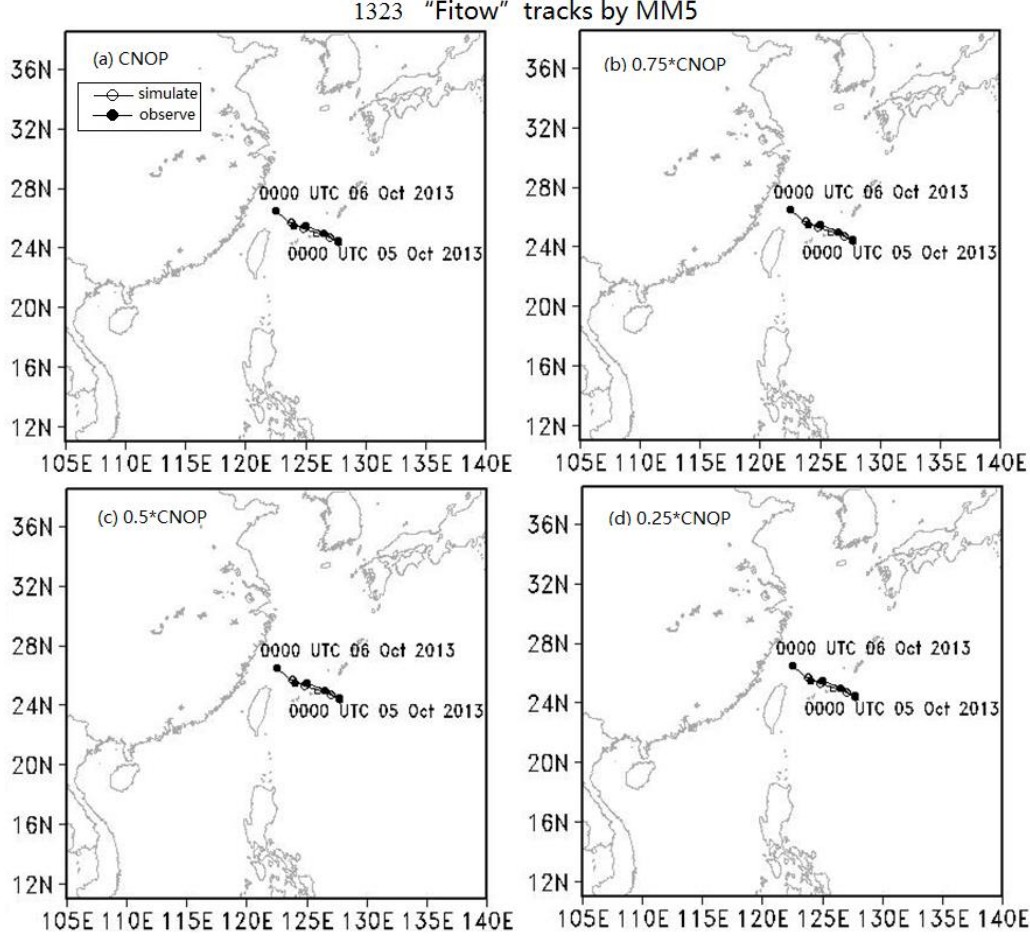

**Figure 13: Simulated TC tracks of MM5 with inserting CNOP or W×CNOP into the initial state in the entire domain for TC Fitow. Solid circles represent observed TC tracks of CMA, and hollow circles show the simulated TC tracks of the MM5 model. (a), (b), (c) and (d) denote CNOP, 0.75×CNOP, 0.5×CNOP and 0.25×CNOP, respectively.**





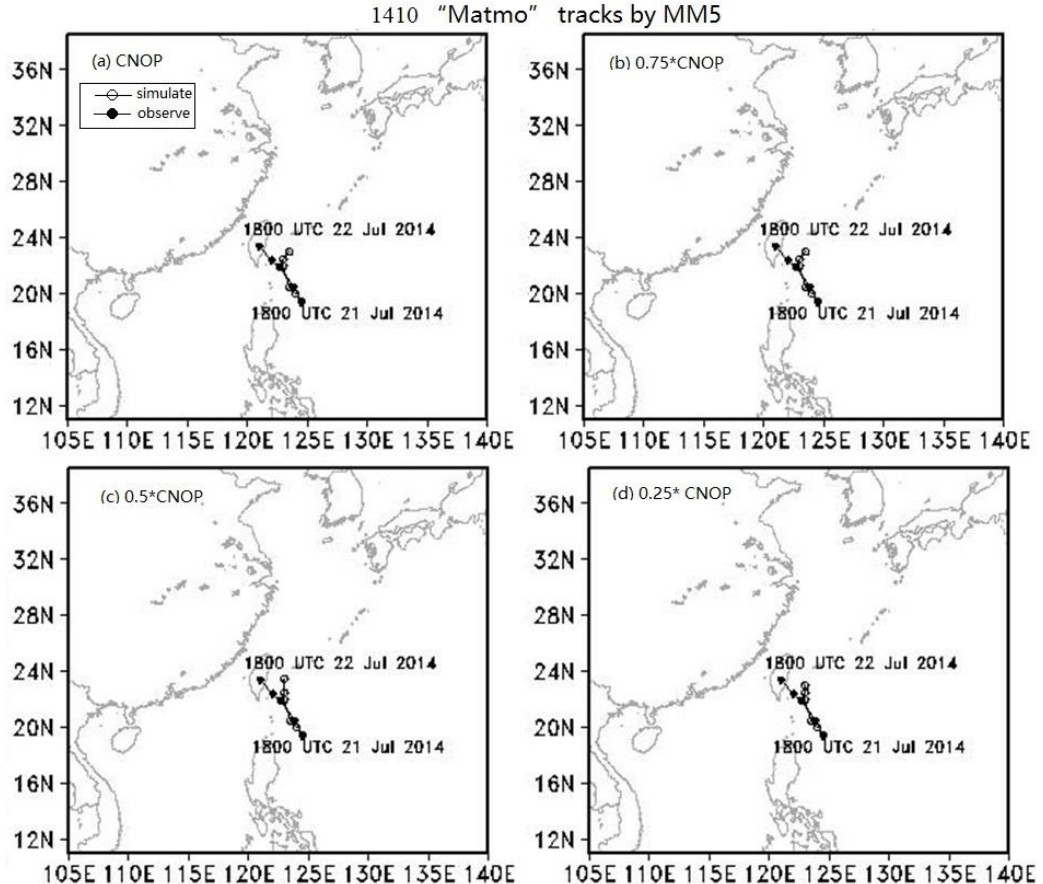

**Figure 14: Simulated TC tracks of MM5 with inserting CNOP or W×CNOP into the initial state in the entire domain for TC Matmo. Solid circles represent observed TC tracks of CMA, and hollow circles show the simulated TC tracks of the MM5 model. (a), (b), (c) and (d) denote CNOP, 0.75×CNOP, 0.5×CNOP and 0.25×CNOP, respectively.**



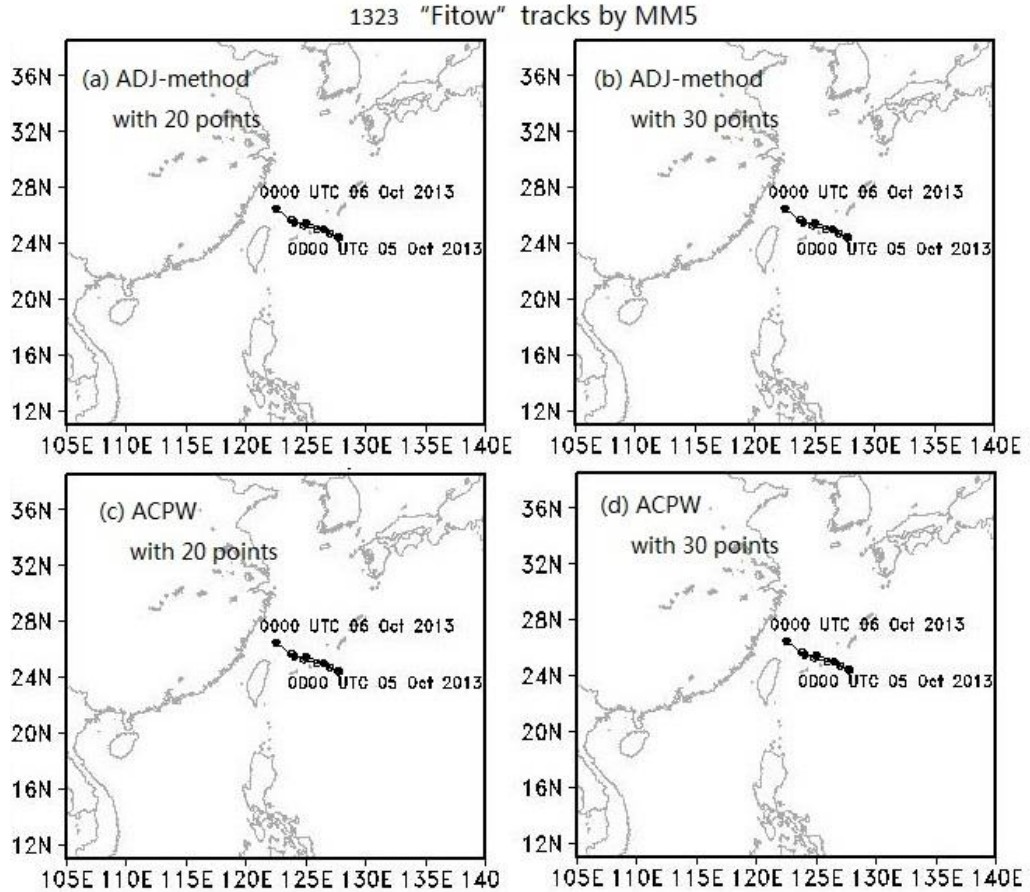

**Figure 15: Simulated TC tracks of MM5 with inserting amended CNOPs, which are reduced to 0.5 time only in the sensitive regions, into the initial state for TC Fitow. Solid circles represent observed TC tracks of CMA, and hollow circles show the simulated TC tracks of the MM5 model. (a), (b), (c) and (d) denote ADJ-method with 20 points, ADJ-method with 30 points, ACPW with 20 points and 30 points, respectively.**




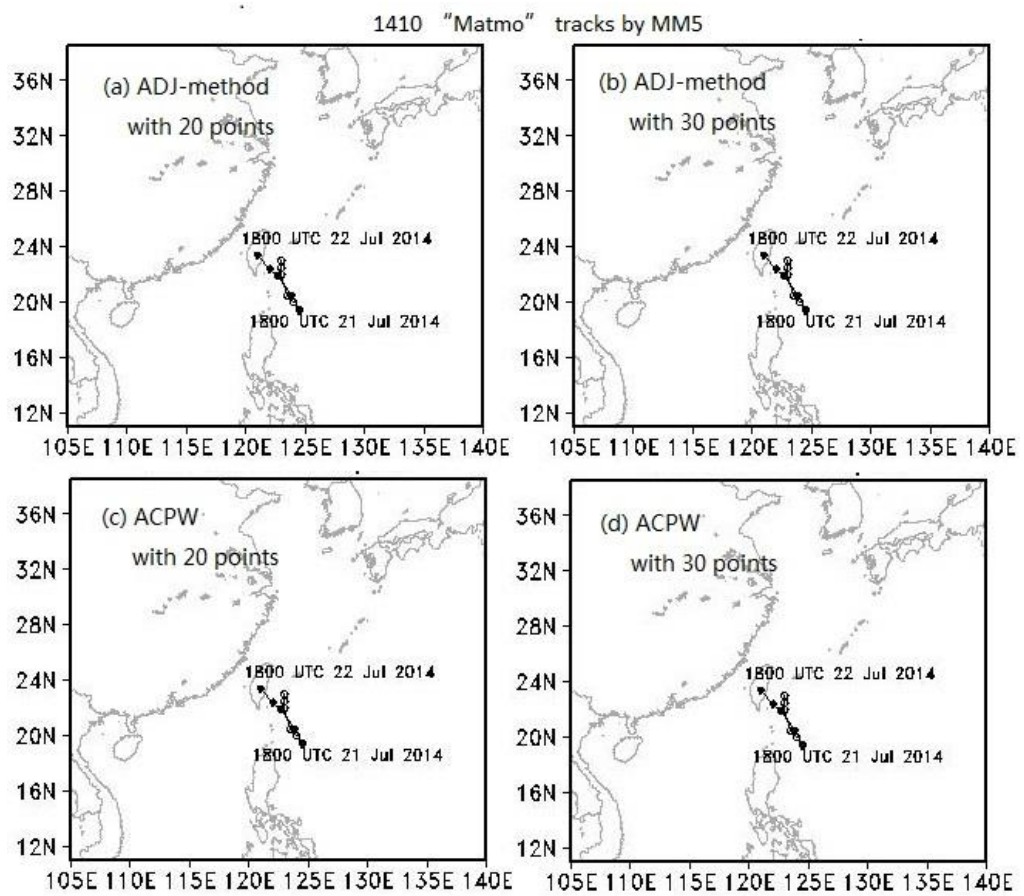

**Figure 16: Simulated TC tracks of MM5 with inserting amended CNOPs, which are reduced to 0.5 time only in the sensitive regions, into the initial state for TC Matmo. Solid circles represent observed TC tracks of CMA, and hollow circles show the simulated TC tracks of the MM5 model. (a), (b), (c) and (d) denote ADJ-method with 20 points, ADJ-method with 30 points, ACPW with 20 points and 30 points, respectively.**

**Table 1: The parameters of ACPW.**

| Name | Meaning | Value |
|---|---|---|
| n | Number of principle components | 50 |
| N | Number of individuals | 420 with 120km<br>200 with 60km |
| a | Adaptive coefficient | Initial: 0.5 |
| ω | Inertia coefficient | 0.8 |
| c1 | Self-awareness to track the historically optimal position | 2.05 |
| c2 | Social-awareness of the particle swarm to track the globally optimal position | 2.05 |



| ϒ | Restraint factor to control the speed | 0.729 |
| θ | Velocity of individual moving | 0.5 |
| r | Local optimizing radius | 8×δ/original dimensions |
| s | Step size of updating individual | 0.6 |
| $p_a$ | Probability of individual escaping from current position | 0.3 |
| Total_Step | The number of iterations | 50 |

**Table 2: The meanings of all symbols**

| Symbols | Values/ components | Meanings |
|---|---|---|
| $\delta u_0$ | $u_0'$, $v_0'$, $T_0'$, $p_{s_0}'$, | Initial perturbation |
| $u_{NT}$ | $u_t'$, $v_t'$, $T_t'$, $p_{s_t}'$ | Nonlinear evolution of perturbed $U_0$ at time t |
| D | Values rely on cases | Verification area |
| σ | (0, 1] | Vertical coordinate |
| $c_p$ | 1005.7 J kg−1 K−1 | Specific heat at constant pressure |
| $R_a$ | 287.04 J kg−1K−1 | Gas constant of dry air |
| $T_r$ | 270K | Constant parameter |
| $p_r$ | 1000hPa | Constant parameter |

**Table 3: The similarities of CNOPs gained from ACPW and ADJ-method.**

| ACPW&ADJ-method | 120km | 60km |
|---|---|---|
| Fitow | -0.83 | 0.43 |
| Matmo | 0.42 | 0.37 |

**Table 4: The ratios of energy for 24-h nonlinear evolutions with inserting CNOPs-gained by ACPW and ADJ-method into the initial states.**

| ACPW/ADJ-method | 120km | 60km |
|---|---|---|
| Fitow | 94.1% | 85.1% |
| Matmo | 87.3% | 70.2% |

5  **Table 5: Benefits (percent, %) gained from reducing CNOPs to 0.5 time in the sensitive regions identified by ADJ-method and ACPW with 20 points.**

| Cases | Methods | 60km | 120km |
|---|---|---|---|
| Fitow | ADJ-method | 3% | 5.93% |
| | ACPW | -0.84% | **8.05%** |





| | | | |
|---|---|---|---|
| Matmo | ADJ-method | 6.12% | 20.90% |
| | ACPW | **20.48%** | 16.26% |

**Table 6: Benefits (percent, %) gained from reducing CNOPs to 0.5 time in the sensitive regions identified by ADJ-method and ACPW with 6 in 120km resolution and 30 points in 60km resolution.**

| Cases | Methods | 60km (30 points) | 120km (6 points) |
|---|---|---|---|
| Fitow | ADJ-method | 3.9 | 1.72% |
| | ACPW | **4.23%** | 0.01% |
| Matmo | ADJ-method | 1.21% | 13.24% |
| | ACPW | **9.75%** | 6.86% |

**Table 7: The time consumption of ADJ-method and ACPW (unit: minutes).**

| Methods | 60km | 120km |
|---|---|---|
| ADJ-method (1)[1] | 79.9 | 12.4 |
| ADJ-method (4)[1] | 321.1 | 49.7 |
| ACPW | **20.8** | **2.74** |