# Peer review of "A Novel Approach for Solving CNOP and its Application in Identifying Sensitive Regions of Tropical Cyclone Adaptive Observations"

_Nonlinear Processes in Geophysics, 2018_

## Referee Comment (RC1) · Anonymous Referee #1 · 18 May 2018

General comment The authors used an approach (ACPW), combining the PSO and the WSA, to solve reduction conditional nonlinear optimal perturbations (CNOPs), then applied them (reduction CNOPs) to identify the sensitive regions of TC adaptive observations. The validity of ACPW were also verified by comparing the similarity of the reduction CNOP yielded by ACPW with the CNOP produced by the traditional optimization algorithm based on the adjoint model. Considering the applications of CNOP, it is an interesting work.

Major remarks The blend optimization algorithm ACPW combines two intelligent algorithms, the PSO and the WSA, to capture the CNOP associated with TC adaptive

observations. Please state the advantages of the both PSO and SWA algorithms, and their performance difference in detail, so that readers can know the motivation that you combine them to coevolve to solve the CNOP. As we know that the largest challenge in intelligent algorithms is to solve high dimension optimization problems. However, the authors, following their previous works (such as Mu et al., 2015a, b), have reduced solving high dimensional CNOPs to an optimization problem in a lower dimensional feature space by using the PCA method, with the number of principal components being 50 in their numerical experiments. For such low dimensional optimization problems, both the PSO and the WSA can also obtain the global optimal solutions quickly. Please use statistical method to demonstrate the better optimization performance of ACPW comparing with the PSO and the SWA in perspective of optimization time and accuracy. There is a great difference at the operation rules of the WSA between the standard version given by Rui Tang et al. (2012) and the formula (6) of this study, please make explanation or correction.

Specific remarks: (1) Page 3, line 24, 26: The variants given in the propagation operator M should be uniform. (2) Page 5, line 8∼9: Please state in detail the rule setting adaptive subswarm coefficient a. (3) Page 5, line 17∼19: It is better to delete these three lines since the description is unnecessary.

Please also note the supplement to this comment:
https://www.nonlin-processes-geophys-discuss.net/npg-2018-17/npg-2018-17-RC1-supplement.pdf

---

## Referee Comment (RC2) · Anonymous Referee #2 · 23 May 2018

General comments: The authors study the computation of perturbations with optimal non-linear growth over forecast time determined with the mesoscale meteorological forecast model MM5 in the context of tropical cyclone forecast. The underlying computational problem is a very high dimensional global minimisation problem. In order to find viable alternatives for using an adjoint, the authors test a combination of two other search algorithms, "particle swarm optimisation" and "wolf search" on a reduced dimension state space with 50 dimension and test their performance against a reference method called "the ADJ method". However, it does not become clear, whether this reference method is used to solve the same problem, which should give identical results provided that all methods find the global minimum. Instead, the optimal perturbations

found are quite different. Also, solving a 50-dimensional problem with 200 (resp. 420; see swarm size from table 1) model integrations at each solver step in 20 to 30 steps (Fig. 2) does not look like a dramatic improvement over conventional methods, and no direct comparison to those is offered. "The ADJ method" is used as a benchmark, but it is ambiguously defined and no attempts on parallelisation are made, not even in the case of multiple starting points, which supposedly can be parallelised trivially. Also the article leaves the impression that "the ADJ method" is run on the full state space, rather than the 50 dimensional PC space. In summary, the comparisons in terms of computational performance are not convincing.

The experiments with the reduced amplitude CNOPs are hard to follow. I had difficulties to understand section 4.3., which is the motivation for the verification and forecast experiments.

Specific comments: In the presentation of the resulting CNOPs, the surface pressure patterns are neither shown nor discussed. No information on the vertical structure of the CNOPs is given. Moisture, an important energy source for tropical cyclones, is not included in the state vector and no justification for this omission is given. The authors do not address the the role of the fixed PC space dimension (and basis?) when comparing patterns at different resolutions. No information on how the excitation of numerical modes is avoided, both in the computation of the CNOPs and when making perturbed forecasts.

Technical comments: Many formulations in the abstract and the article are confusing on a language level, to name only a few: "...suggest that the use of an ocean coupled model needs to be conscious,..." (page 2, line 13), "the mutual affection of binary typhoons" (page 2 line 14), "[wolf search] ... takes long consuming time." (page 4, line 6). Language editing is encouraged.

page 5, formula 6: What is the update for $u\_i$ if neither of the two conditions is satisfied?

page 8, formula 10: Is this using the same energy norm as formula 10? If not, how are

the different variables combined?

---

## Author Response (AR1)

**List of Responses**

Responds to the Anonymous Referee #1's comments:

Special thanks for your good comments which are very useful for us to improve the paper.

1. Response to comment: Please state the advantages of the both PSO and WSA algorithms, and their performance difference in detail, so that readers can know the motivation that you combine them to coevolve to solve the CNOP. Please use statistical method to demonstrate the better optimization performance of ACPW comparing with the PSO and the WSA in perspective of optimization time and accuracy.

Response: It is really true as Rreview1 suggested that we need to clarify the advantages of the both PSO and WSA algorithms and analyze the the better optimization performance of ACPW. Therefore we have illustrated this in the Section 4.1.

"To evaluate the advantages of the ACPW algorithm, we run the PSO, WSA and ACPW programs 10 times and then compare the maximum, minimum and mean objective values as well as the RMSE.

4.1 The advantages of the ACPW algorithm

Because the statistical analysis results are similar for the two TCs with the two resolutions, we only describe the analysis of Fitow at a resolution of 60 km. Table 3 presents the maximum objective value, the minimum objective value, the mean objective value and the RMSE of the 10 results.

Table 3: The analysis results of the PSO, WSA and ACPW methods.

| Algorithm | Maximum Value | Minimum Value | Mean Value | RMSE |
|-----------|---------------|---------------|------------|------|
| PSO | 1034.192573 | 724.086002 | 900.7488578 | **0.121400896** |
| WSA | 1628.841294 | 323.7493169 | 930.9103862 | 0.431193448 |
| ACPW | **2240.275956** | 1243.377921 | **1542.505251** | 0.216750584 |

In Table 3, the maximum objective value is gained from the ACPW algorithm, and its mean value is also more than the other two algorithms. However, the RMSE of PSO is the smallest, which shows the best stability.

For additional analysis, we draw a box-plot of the 10 results for the PSO, WSA and ACPW algorithms, as shown in Fig. 3.

[Figure]

**Figure 3: Box-plot of the PSO, WSA and ACPW methods for TC Fitow at 60 km resolution. The red box denotes PSO, the green box is for the WSA, and the blue box shows the results of the ACPW algorithm.**

PSO has the narrowest range of values, although the objective values are smaller than the other two algorithms. The WSA has the widest range of values, although the objective values are also smaller than the ACPW algorithm. The ACPW algorithm has the second-best stability, although it has the best objective values. The experiments display the stability of PSO and the exploitation of the WSA. We combine the advantages of them and develop the ACPW algorithm to solve CNOPs. The analysis results demonstrate that the hybrid strategy and cooperation co-evolution is useful and effective."

2. Response to comment: There is a great difference at the operation rules of the WSA between the standard version given by Rui Tang et al. (2012) and the formula (6) of this study, please make explanation or correction.

Response: We are very sorry about errors in this paper and have corrected them in Page 5, line 2-9. "

$$\begin{cases} u_i^{k+1} = u_i^k + \theta \cdot r \cdot rand(\ ) & Prey \\ u_i^{k+1} = u_i^k + \theta \cdot s \cdot escape(\ ) & Escape \end{cases} \tag{6}$$

where the superscript $k$ or $k+1$ is also the iterative step, $\theta$ is the velocity, $r$ is the local optimizing radius, which is smaller than the global constraint radius $\delta$, $rand(\ )$ is the random function, whose mean value is distributed in [-1,1], $escape(\ )$ is the function for calculating a random position, which is 3 times larger than $r$, and $s$ is the step size of the updating individual.

As described in Eq. (6), the wolf has two behaviours, i.e., prey and escape. The prey behaviour uses the first sub-formula, and the second one is for the escape function, which happens in every iteration when

the condition $p > p_a$ is satisfied, where $p$ is a random number in [0,1], and $p_a$ is the probability of individual escaping from the current position. "

3. Response to comment: (1) Page 3, line 24, 26: The variants given in the propagation operator M should be uniform.
Response: As Rreview1 suggested that we rewritten this part in Page 3, line 25.

"$U_t = M_{t_0 \to t}(U_0)$"

4. Response to comment: (2) Page 5, line 8-9: Please state in detail the rule setting adaptive subswarm coefficient a.

Response: As Rreview1 suggested that we have added the rule setting adaptive subswarm coefficient a in Page 5, line13-16.

"$\alpha = \begin{cases} \alpha + 0.05, & if \quad the \ bestvalue - \ current \ value < \ \varepsilon \\ \alpha - 0.05, & else \end{cases}$

In this paper, before we update the individuals, $\alpha$ is calculated, and then we divide the entire initial swarm into two subswarms according to the $\alpha$ value, i.e., the number of individuals depending on the PSO's rule is $\alpha \times N$, and the other number is $(1 - \alpha) \times N$. We set the initial value of $\varepsilon$ and $\alpha$ to 0.1 and 0.5, respectively. "

5. Response to comment: (3) Page 5, line 17-19: It is better to delete these three lines since the description is unnecessary.

Response: We need to explain about this part. The reason for writing this part is to present the performance of our algorithms in this paper under those computer hardware environments. If the reader needs to compare with our results, they should have the same environments. Hence, we did not delete them.

In addition, we have improved the quality of our manuscript by American Journal Experts editing service and tracked the changes using revisions in the manuscript 'Revised Manuscript with Track Changes'.

**List of Responses**

Responds to the Anonymous Referee #2's comments:

Special thanks for your good comments which are very useful for us to improve the paper.

1. Response to comment: In order to find viable alternatives for using an adjoint, the authors test a combination of two other search algorithms, "particle swarm optimisation" and "wolf search" on a reduced dimension state space with 50 dimension and test their performance against a reference method called "the ADJ method". However, it does not become clear, whether this reference method is used to solve the same problem, which should give identical results provided that all methods find the global minimum. Also, solving a 50-dimensional problem with 200 (resp. 420; see swarm size from table 1) model integrations at each solver step in 20 to 30 steps (Fig. 2) does not look like a dramatic improvement over conventional methods, and no direct comparison to those is offered.

Response: It is really true as Rreview2 suggested that we should give identical results provided that all methods find the global minimum. And we run the PSO, WSA and ACPW programs 10 times and then compare their results. It is commonly known that all intelligent algorithms are stochastic; that is, even when the input is the same in different trials, the output may be different. Hence, it hard to obtain the same result. But we can use the maximum, minimum and mean objective values as well as the RMSE to evaluate the algorithm. Therefore, we have illustrated this in the Section 4.1.

"To evaluate the advantages of the ACPW algorithm, we run the PSO, WSA and ACPW programs 10 times and then compare the maximum, minimum and mean objective values as well as the RMSE.

4.1 The advantages of the ACPW algorithm

Because the statistical analysis results are similar for the two TCs with the two resolutions, we only describe the analysis of Fitow at a resolution of 60 km. Table 3 presents the maximum objective value, the minimum objective value, the mean objective value and the RMSE of the 10 results.

Table 3: The analysis results of the PSO, WSA and ACPW methods.

| Algorithm | Maximum Value | Minimum Value | Mean Value | RMSE |
|---|---|---|---|---|
| PSO | 1034.192573 | 724.086002 | 900.7488578 | **0.121400896** |
| WSA | 1628.841294 | 323.7493169 | 930.9103862 | 0.431193448 |

| | | | | |
|---|---|---|---|---|
| ACPW | **2240.275956** | 1243.377921 | **1542.505251** | 0.216750584 |

In Table 3, the maximum objective value is gained from the ACPW algorithm, and its mean value is also more than the other two algorithms. However, the RMSE of PSO is the smallest, which shows the best stability.

For additional analysis, we draw a box-plot of the 10 results for the PSO, WSA and ACPW algorithms, as shown in Fig. 3.

[Figure]

**Figure 3: Box-plot of the PSO, WSA and ACPW methods for TC Fitow at 60 km resolution. The red box denotes PSO, the green box is for the WSA, and the blue box shows the results of the ACPW algorithm.**

PSO has the narrowest range of values, although the objective values are smaller than the other two algorithms. The WSA has the widest range of values, although the objective values are also smaller than the ACPW algorithm. The ACPW algorithm has the second-best stability, although it has the best objective values. The experiments display the stability of PSO and the exploitation of the WSA. We combine the advantages of them and develop the ACPW algorithm to solve CNOPs. The analysis results demonstrate that the hybrid strategy and cooperation co-evolution is useful and effective."

2. Response to comment: **"The ADJ method"** is used as a benchmark, but it is ambiguously defined and no attempts on parallelisation are made, not even in, the case of multiple starting points, which supposedly can be parallelised trivially. Also the article leaves the impression that "the ADJ method" is run on the full state space, rather than the 50 dimensional PC space. In summary, the comparisons in terms of computational performance are not convincing.

Response: As Rreview2 suggested that we inserted the reference of the ADJ-method in L5-6, p.3. "Specific details of the ADJ-method can be found in Zhou (2009)."

As Rreview2 mentioned that the multiple starting points can be paralleled, but the time consumption will not be less than using one starting point under the same computer hardware environments. When we analyze the efficiency of the ACPW algorithm in Section 4.5, the ADJ-method using one initial guess field (starting point) is compared with the ACPW algorithm. And the speedup of the ACPW reaches 4.53 and 3.84 for the different resolutions.

"To promote the efficiency of the ACPW algorithm, we parallelize it with MPI technology. The time consumption of each case is nearly the same. Hence, we can use one group of experimental results to elucidate the efficiency of the ACPW algorithm. Because the ADJ-method cannot be parallelized because each input depends on the output of the previous step, its time consumption is not changed. Moreover, because this method generally uses 4~8 initial guess fields to obtain the optimal value, we use one and four initial first guess fields to determine the CNOPs. The time consumptions of the ADJ-method and ACPW algorithm are shown in Table 8.

Table8: The time consumption of the ADJ-method and ACPW algorithm (unit: minutes).

| Methods | 60 km | 120 km |
|---|---|---|
| ADJ-method (1)[1] | 79.9 | 12.4 |
| ADJ-method (4)[1] | 321.1 | 49.7 |
| ACPW | **20.8** | **2.74** |

1. ADJ-method (1) means using 1 initial guess field and ADJ-method (4) means using 4 initial guess fields.

At 120 km resolution, the time consumptions of the ADJ-method using 1 and 4 initial guess fields are 12.4 minutes and 49.7 minutes, respectively. At 60 km resolution, the time consumptions are 79.9 minutes and 321.1 minutes, respectively. Unlike the ADJ-method, the ACPW algorithm can be parallelized. When using 22 cores, the ACPW method requires much less time, i.e., 2.74 minutes at 120 km resolution and 20.8 minutes at 60 km resolution. Obviously, the ACPW has higher efficiency. Compared to the ADJ-method (1), the speedup reaches 4.53 and 3.84 for the different resolutions. Compared to the ADJ method (4), the speedup reaches 18.14 and 15.44. Although the different initial guess fields are calculated in parallel, the time consumption must be more than for the ADJ-method (1); the ACPW algorithm is also faster than the ADJ-method.
"

In addition, when the ACPW algorithm calculated the objective value, we use the nonlinear model on the full state space, only update the individual with the 50 dimensions.

3. Response to comment: The experiments with the reduced amplitude CNOPs are hard to follow. I had difficulties to understand section 4.3., which is the motivation for the verification and forecast experiments.

Response: For the Section 4.3, we want to investigate the validity of the sensitive regions identified using CNOPs, and we have two assumptions:

"When adding adaptive observations in sensitive regions, the surrounding environment is idealized, and the improvements from adding observations reduces the original errors by a factor of 0.5.
The obtained CNOPs can be seen as the optimal initial perturbations. Once we reduce them in the

Therefore, we design two groups of idealized experiments. CNOPs are optimal initial perturbations having the maximum nonlinear evolutions at the forecast time. Under these assumptions, by reducing the CNOPs to W×CNOPs and inserting them into the initial states we can investigate how the reductions in the CNOPs influence TC forecast skill.

4. Response to comment: In the presentation of the resulting CNOPs, the surface pressure patterns are neither shown nor discussed. No information on the vertical structure of the CNOPs is given. Moisture, an important energy source for tropical cyclones, is not included in the state vector and no justification for this omission is given. The authors do not address the the role of the fixed PC space dimension (and basis?) when comparing patterns at different resolutions. No information on how the excitation of numerical modes is avoided, both in the computation of the CNOPs and when making perturbed forecasts.

Response:
As Review2 mentioned that we did not discuss the surface pressure patterns and the vertical structure of the CNOPs, because the purpose of this paper is to identify the adaptive observation sensitive areas, we follow the study of Dr, Zhou that the total dry energy have higher benefits than other strategies (Zhou and Zhang, 2014). Therefore, the information of the surface pressure patterns and the vertical structure of the CNOPs are contained in the total dry energy. In addition, Dr. Zhou has proved that the sensitive regions gained by the dry energy have higher benefits than those obtained from the moist energy (Zhou, 2009). In this paper, we only considered the total dry energy.
For the question that "The authors do not address the role of the fixed PC space dimension (and basis?) when comparing patterns at different resolutions", the numbers of PCs in this paper are determined by the many experiments, and the analysis of the different numbers are plotted in our previous studies.
Finally, in this paper, we also use the nonlinear model, but avoid using the adjoint model to calculate the gradient.

5. Response to comment: Many formulations in the abstract and the article are confusing on a language level, to name only a few: "...suggest that the use of an ocean coupled model needs to be conscious,..." (page 2, line 13), "the mutual affection of binary typhoons" (page 2 line 14), "[wolf search] ... takes long consuming time." (page 4, line 6). Language editing is encouraged.

Response: As Review2 suggested that we have improved the quality of our manuscript by American Journal Experts editing service and tracked the changes using

revisions in the manuscript 'Revised Manuscript with Track Changes'.

6. Response to comment: What is the update for ui if neither of the two conditions is satisfied?

Response: We are very sorry about errors in this paper and have corrected them in L2-9 Page 5. "

$$\begin{cases} u_i^{k+1} = u_i^k + \theta \cdot r \cdot rand(\ ) & Prey \\ u_i^{k+1} = u_i^k + \theta \cdot s \cdot escape(\ ) & Escape \end{cases} \quad (6)$$

where the superscript $k$ or $k+1$ is also the iterative step, $\theta$ is the velocity, $r$ is the local optimizing radius, which is smaller than the global constraint radius $\delta$, $rand(\ )$ is the random function, whose mean value is distributed in [-1,1], $escape(\ )$ is the function for calculating a random position, which is 3 times larger than $r$, and $s$ is the step size of the updating individual.

As described in Eq. (6), the wolf has two behaviours, i.e., prey and escape. The prey behaviour uses the first sub-formula, and the second one is for the escape function, which happens in every iteration when the condition $p > p_a$ is satisfied, where $p$ is a random number in [0,1], and $p_a$ is the probability of individual escaping from the current position. "

7. Response to comment: page 8, formula 10: Is this using the same energy norm as formula 10? If not, how are the different variables combined?

Response: formula (10) is used to calculate the similarity between the CNOPs, every CNOP has the same components, so we did not use the norm. Actually, the formula is for solving the Cosine similarity.

[revised manuscript text omitted]

---

## Referee Report (RR1)

Major remarks

- Here are some questions on the additional numerical experiments shown in Fig. 3 (Box-plot of the PSO, WSA and ACPW methods for TC Fitow at 60 km resolution) of the revised manuscript. In Fig.2, we find large differences among the initial objective function values for the PSO, WSA and ACPW methods, and the one for the ACPW method is even approximately equal to the final objective function value obtained by the PSO method. Does this disparity exists in the numerical experiments shown in Fig. 3? If so, the comparison of the objective function values gained respectively by the PSO, WSA and ACPW methods cannot support the better optimization performance of the ACPW methods. Accordingly, the descending degree of the objective function value for the three methods should be compared.

---

## Author Response (AR2)

**List of Responses**

Responds to the Anonymous Referee #1's comments:

Special thanks for your good comments which are very useful for us to improve the paper.

1. Response to comment: Here are some questions on the additional numerical experiments shown in Fig. 3 (Box-plot of the PSO, WSA and ACPW methods for TC Fitow at 60 km resolution) of the revised manuscript. In Fig.2, we find large differences among the initial objective function values for the PSO, WSA and ACPW methods, and the one for the ACPW method is even approximately equal to the final objective function value obtained by the PSO method. Does this disparity exists in the numerical experiments shown in Fig. 3? If so, the comparison of the objective function values gained respectively by the PSO, WSA and ACPW methods cannot support the better optimization performance of the ACPW methods. Accordingly, the descending degree of the objective function value for the three methods should be compared.

Response: As Reviewer mentioned that the Fig. 2 is unsuitable in the revised manuscript. Actually, the purpose of using Fig. 2 is to illustrate the different performance of the PSO, WSA and ACPW methods, but we only use one group experiments. Hence, we delete it in the new revised manuscript.

And to answer the question "Does this disparity exists in the numerical experiments shown in Fig. 3?", we did the statistical analysis of the first objective values, and plotted their distributions of the different methods in the new Fig. 3 in line 9-18, P. 8.

" Since these three algorithms are all heuristic algorithms with randomness, and the initial inputs are generated by random way, the initial objective value is different for every running. To analyse the effect of initial objective values on the different algorithms, we exhibit the objective value scope of the PSO, WSA and ACPW algorithms after the first iteration in Fig. 3.

[Figure]

**Figure 3: The first objective value scope of the PSO, WSA and ACPW methods. PSO is denoted as**

**the red line, the WSA is shown as the green line, and the ACPW algorithm is represented as the blue line.**

In Fig. 3, for convenience, only integer is indicated in the coordinate system. In 10 experiments, the PSO has the narrowest scope from 467.1719 to 781.6482. The WSA and ACPW algorithms have similar value spans, which are wider than the PSO, but the objective values of the ACPW are higher. And the value scope is reasonable according to the characteristic of these three algorithms. The WSA has the strongest randomness, the PSO is the most stable, and the ACPW combines the advantages of the two.    From the results, we cannot find the direct relationship between the initial objective value and the final result, but a better first objective value has a beneficial effect on finding the optimal value."

As Reviewer suggested that we compared the changing degree of the objective function value for the three algorithms to illustrate the performance of the different algorithms, and the results were shown in the new Fig. 4 in line 19-29, P. 8.

[revised manuscript text omitted]